# Observation of Kekulé vortices around hydrogen adatoms in graphene

Yifei Guan [1], Clement Dutreix [2], Héctor González-Herrero [3,4], Miguel M. Ugeda [5,6,7], Ivan Brihuega [3,4,8] ✉, Mikhail I. Katsnelson [9], Oleg V. Yazyev [1] ✉ & Vincent T. Renard [10] ✉

Fractional charges are one of the wonders of the fractional quantum Hall effect. Such objects are also anticipated in two-dimensional hexagonal lattices under time reversal symmetry—emerging as bound states of a rotating bond texture called a Kekulé vortex. However, the physical mechanisms inducing such topological defects remain elusive, preventing experimental realization. Here, we report the observation of Kekulé vortices in the local density of states of graphene under time reversal symmetry. The vortices result from intervalley scattering on chemisorbed hydrogen adatoms. We uncover that their $2\pi$ winding is reminiscent of the Berry phase $\pi$ of the massless Dirac electrons. We can also induce a Kekulé pattern without vortices by creating point scatterers such as divacancies, which break different point symmetries. Our local-probe study thus confirms point defects as versatile building blocks for Kekulé engineering of graphene's electronic structure.

Real-space topological defects in crystals exhibit exotic electronic properties[1,2], especially when combined with the reciprocal-space topological phase hosted by the bulk[3,4]. In two-dimensional hexagonal lattices, a vortex in the Kekulé order parameter is of particular interest for charge fractionalization without breaking time-reversal symmetry[5,6]. The Kekulé pattern in graphene corresponds to the $\sqrt{3} \times \sqrt{3}R30°$ unit cell tripling, with a distinct bond order within one of the three equivalent hexagonal rings. These three degenerate states define an angular order parameter space as shown in Fig. 1a. A Kekulé vortex of winding $2\pi$ corresponds to the alternation of the three Kekulé domains upon encircling a singularity, which could be implemented in optical[7,8] and acoustical[9,10] metamaterials. In graphene, recent experiments have demonstrated that a Kekulé bond texture emerges from the intervalley coherent quantum Hall ferromagnet states in the zeroth Landau level[11–13]. These states can host skyrmionic topological excitations appearing as Kekulé vortices[13–15].

However, these schemes require a strong magnetic field and the Kekulé vortex, without breaking time-reversal symmetry, remains out of reach. At zero magnetic field, theory shows that a missing electronic site provides a fractionalization mechanism analogous to that of the Kekulé vortex.[16,17] Such point defects can scatter electrons from one valley to another, thereby connecting the two valleys and offering a way to stabilize the Kekulé bond texture[18–24]. We now demonstrate that they also induce Kekulé vortices in graphene.

## Results

### Observation of a Kekulé vortex near an H atom on graphene

Figure 1b shows a scanning tunneling microscopy (STM) image of graphene with a single chemisorbed hydrogen adatom (see Methods for experimental details and Supplementary Fig. 1 for another example). Near the H atom, a strong quasi-particle interference (QPI) signal is seen. It reveals a hexagonal superlattice commensurate with that of

[1]Institute of Physics, École Polytechnique Fédérale de Lausanne (EPFL), CH-1015 Lausanne, Switzerland. [2]Univ. Bordeaux, CNRS, LOMA, UMR 5798, F-33400 Talence, France. [3]Departamento de Física de la Materia Condensada, Universidad Autónoma de Madrid, E-28049 Madrid, Spain. [4]Condensed Matter Physics Center (IFIMAC), Universidad Autónoma de Madrid, E-28049 Madrid, Spain. [5]Donostia International Physics Center (DIPC), Paseo Manuel de Lardizábal 4, 20018 San Sebastián, Spain. [6]Centro de Física de Materiales (CSIC-UPV-EHU), Paseo Manuel de Lardizábal 5, 20018 San Sebastián, Spain. [7]Ikerbasque, Basque Foundation for Science, 48013 Bilbao, Spain. [8]Instituto Nicolás Cabrera, Universidad Autónoma de Madrid, E-28049 Madrid, Spain. [9]Institute for Molecules and Materials, Radboud University, Heijendaalseweg 135, 6525AJ Nijmegen, The Netherlands. [10]Univ. Grenoble Alpes, CEA, Grenoble INP, IRIG, PHELIQS, 38000 Grenoble, France. ✉e-mail: ivan.brihuega@uam.es; oleg.yazyev@epfl.ch; vincent.renard@cea.fr

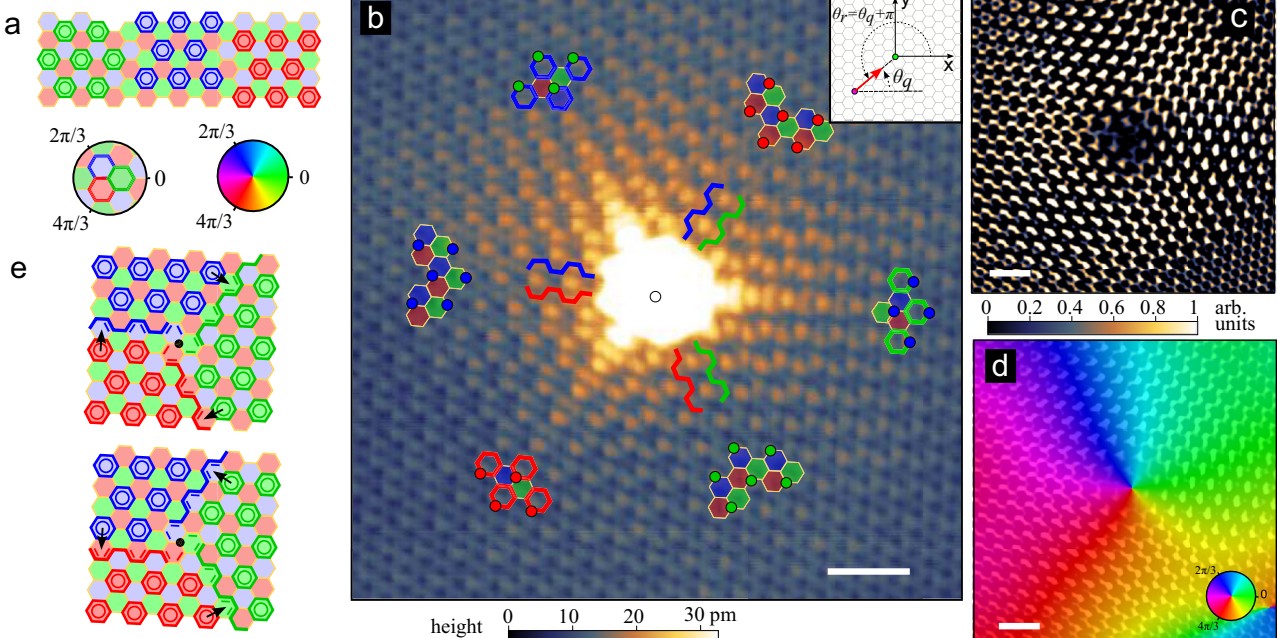

**Fig. 1 | Observation of the Kekulé vortex induced by a hydrogen adatom on graphene. a** Schematic illustration showing three distinct Kekulé orders in graphene. **b** STM image of graphene with a hydrogen adatom in the center for a tip bias $V_b$=400 mV and tunneling current $i_t$ = 45.5 pA. The scale bar is 1 nm. The colored tiling evidences three domains, each corresponding to one of the three Kekulé orders (bold hexagons) and separated from one another by domain walls along the armchair direction. The pseudo-spin of incoming electrons scattering off the H atom is locked on the azimutal coordinate $\theta_r$ of the STM tip[25] (upper right inset). **c** Kekulé bond-texture signal extracted according to the methodology described in the main text and the Supplementary information. The graphene lattice signal is included to highlight the donut-like patterns in each domain. The bare signal is plotted in Fig. 2a. **d** Phase of the Kekulé signal shown in **c** with the Kekulé order parameter defined by the color wheel in **a**. The corresponding signal is overlaid as a guide for the eyes. The scale bars in panels **c** and **d** are 1 nm. **e** Clar's sextet configurations of graphene in the presence of a hydrogen adatom (black dot) illustrating the emergence of a Kekulé vortex.

graphene but with a unit cell three times larger, as emphasised by the three-colour tiling (See Supplementary Fig. 2 for details). The pattern is dominated by the onsite (i.e. on-carbon atom) bright dots on sublattice B (assuming the H adatom is on sublattice A), which alternates between the three nonequivalent B sites. The coloured dots Fig. 1b highlight that this onsite signal winds $4\pi$ when circling around the adatom, consistent with previous studies[25–27]. A closer inspection of the interference pattern also reveals a strong donut-like signal on some benzene rings. We highlight them with bold-coloured hexagons in Fig. 1b. This color tiling allows us to identify three distinct Kekulé domains separated from one another by domain walls along the armchair directions. This indicates that the Kekulé bond texture winds $2\pi$ when circling around the adatom.

To further confirm the existence of the $2\pi$ vortex, we extract the bond-centered signal directly from the STM data, following the approach of ref. 28. In particular, we use a modified Geometrical Phase Analysis[29], in which we exploit the fact that for intervalley scattering, the sublattice A, sublattice B, and bond-centered contributions each transform under different irreducible representations of the threefold symmetry (see Supplementary Information). The result is shown in Fig. 1c, d. The images respectively present the magnitude and the phase of the bond signal for intervalley scattering, which defines the Kekulé order parameter. While the intensity shows a Kekulé texture, the phase exhibits a vortex that winds $2\pi$ around the H adatom. This data analysis demonstrates that the chemisorbed H adatom induces a Kekulé vortex on the surrounding graphene bonds.

Interestingly, the Kekulé vortex supports an interpretation in terms of Clar's sextet theory[30], a set of rules explaining the stability of aromatic molecules in chemistry. A sextet represents the six resonantly delocalized $\pi$ electrons by a circle. Clar's rules state that adjacent hexagons cannot be aromatic sextets simultaneously, and that the most stable bond configuration maximizes the number of Clar's

sextets. Graphene admits three equivalent resonant Clar configurations corresponding to three Kekulé orders (Fig. 1a)[31]. The hydrogen adatom effectively removes one site from the lattice, and the circulation of delocalized electrons in the adjacent benzene rings is obstructed. This lifts the degeneracy of Clar's resonant configurations allowed in its surrounding. We then find two possible configurations shown in Fig. 1e. The only freedom lies in the positions of the double covalent bonds along the boundaries between two Kekulé domains. Thus, Clar's sextet representation is consistent with Fig. 1c and supports our observation of the Kekulé vortex in graphene. In Supplementary Information, we formalize the relation between these two pictures.

## Establishing the electronic origin of the Kekulé vortex

We now show that the Kekulé vortex has a purely electronic origin reminiscent of graphene's band topology. The bond texture of the Kekulé type relates to the electron density between carbon atoms. Intervalley scattering resolved at the tip position **r** yields the following bond contribution to the local density of states (LDOS):

$$\Delta\rho_{bond}(\mathbf{r}) \propto \sum_{n=1,2,3} \sin(\Delta\mathbf{K}_n \cdot \mathbf{r} - \theta_r + \phi_n), \qquad (1)$$

where $\phi_n = (2n+1)\pi/3$ and $\Delta\mathbf{K}_n$ is a scattering wavevector between the valleys responsible for the hexagonal superlattice pattern (see Supplementary Information). This bond contribution can be understood as a two-path loop interference allowed by the overlap of neighbouring $p_z$ orbitals of carbon atoms.

The Kekulé bond texture originates from the polar angle $\theta_r$ indexing the STM tip orientation around the adatom. Indeed, in such QPI signal which is dominated by backscattering, the signal measured at the tip position is determined by the interference of incoming

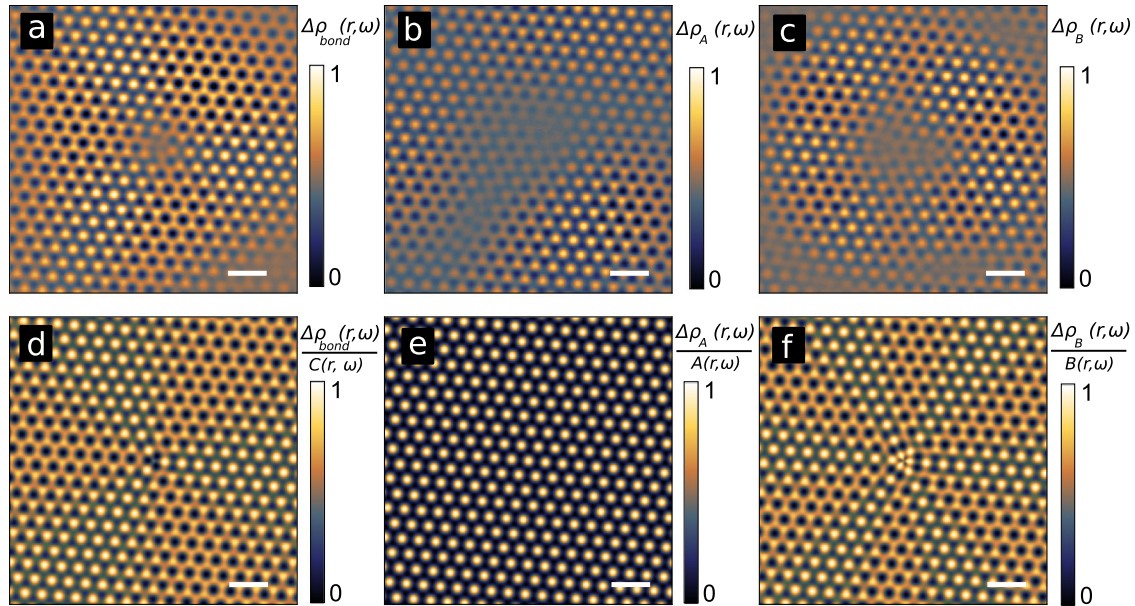

**Fig. 2 | Contributions to the intervalley scattering signal.** Experimental image of $\Delta\rho_{bond}(\mathbf{r})$ (**a**), $\Delta\rho_A(\mathbf{r})$ (**b**) $\Delta\rho_B(\mathbf{r})$ (**c**) from Fig. 1b. The corresponding contributions calculated by the T-matrix approach are shown in **d, e** and **f**. The contributions are normalized by their prefactors for a straightforward comparison (see Supplementary Information). The scale bars are 1 nm.

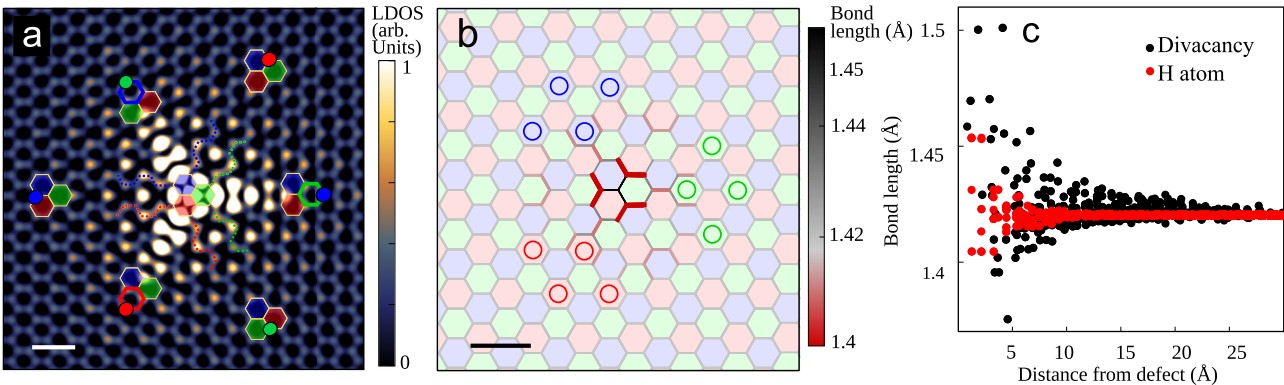

**Fig. 3 | Kekulé vortex induced by the hydrogen adatom from first principles.** **a** STM image simulated using DFT with a tiling defined in the same way as in Fig. 1b. The calculated LDOS was integrated between 0 and 400 meV as in the experiment. The scale bar is 0.5 nm. **b** Relaxed atomic structure of graphene with a hydrogen adatom. The bond lengths are coded as the color and thickness of the bonds. The colored tiling is superposed on the graphene lattice to show the winding of the bond length. The scale bar is 0.5 nm. **c** Calculated bond length distribution as a function of distance to the hydrogen adatom and divacancy defect.

electrons and scattered electrons on the adatom which wavevector orientation $\pm\theta_{\mathbf{q}}$ is determined by the tip position through $\theta_{\mathbf{r}} = \theta_{\mathbf{q}} + \pi$ (see inset in Fig. 1b and ref. 25). The wavevector orientation also determines the wavefunction of the massless relativistic electrons around a Dirac point in momentum space. We assume an incoming wavefunction $|\psi\rangle$ of the form $\sqrt{2}|\psi\rangle = |A\rangle + e^{i\theta_{\mathbf{q}}}|B\rangle$. When cycling along a closed path $\mathcal{C}$ enclosing a Dirac point, the wavefunction gains the Berry phase

$$\varphi_B = i\oint_{\mathcal{C}} \langle\psi|\nabla_{\mathbf{q}}\psi\rangle \cdot d\mathbf{q} = \frac{1}{2}\oint_{\mathcal{C}} d\theta_{\mathbf{q}}. \qquad (2)$$

Due to the lock-in relation $\theta_{\mathbf{r}} = \theta_{\mathbf{q}} + \pi$, cycling the STM tip around the adatom is equivalent to varying the wavevector around a Dirac point in momentum space. Thus, the $2\pi$ vortex on the bonds derives from the topological Berry phase of the scattering wavefunctions.

In addition to the bond contribution above, the usual onsite LDOS modulations $\Delta\rho_A(\mathbf{r})$ and $\Delta\rho_B(\mathbf{r})$ also contribute to the STM signal[25]. All these contributions are compared to the experimental signals in Fig. 2, which shows very good agreement between theory and experiments. (See also Supplementary Information and Supplementary Fig. 3 for more details.) We would like to point out that the experimentally observed vortex is also reproduced by both our tight-binding and density functional theory (DFT) calculations (see Methods, Fig. 3 and Supplementary Fig. 4). We note that the vortices are not affected by the energy integration of the LDOS, since the scattering wave-vectors $\Delta\mathbf{K}_n$ are energy independent. The energy-resolved STM images shown in Supplementary Fig. 5 confirm this property. Thus, our theoretical studies show that the Kekulé vortex we observe in Fig. 1b derives from an intrinsic topological property of the massless Dirac (that is, chiral in pseudospin[32]) wavefunctions scattering on the adatom.

## Discussion

The Kekulé vortices proposed by Hou et al. result from a structural distortion of the bonds and host zero-energy bound states in an excitation gap that are compatible with charge fractionalization

scenarios[5,6]. This raises the question of whether the Kekulé vortex reported here is also accompanied i) by a structural Kekulé distortion and ii) by zero-energy quasi-bound states compatible with fractional excitations.

## Structural relaxation

To investigate whether the Kekulé vortex also exhibits a Kekulé distortion, we perform DFT calculations that include the lattice relaxation effects (see Methods). The simulated STM image (Fig. 3a) reproduces accurately the experiment and, in particular, the $2\pi$ vortex. While the presence of the hydrogen adatom does induce minor lattice distortions that are consistent with the winding of the structural Kekulé distortion (Fig. 3b, c), the results are essentially the same as the ones provided by our analytical description in Eq. (1) and tight-binding calculations (Supplementary Fig. 4) that do not include any lattice distortion effects whatsoever. Furthermore, performing DFT calculations without lattice relaxation does not affect the presence of the vortex (Supplementary Fig. 6). This further confirms the electronic origin of the Kekulé vortex.

## Associated bound state

We now discuss the existence of quasi-bound states associated with the Kekulé vortex. The hydrogen adatom forms a covalent bond with a carbon atom of the graphene lattice. This changes the hybridization of the hydrogenated carbon atom from $sp^2$ to $sp^3$, as in diamond[33,34]. Thus, the carbon atom is essentially unavailable for the $\pi$ electrons and shares similarities with a single-atom ideal vacancy (i.e. without atomic reconstruction). The simplest description mainly consists of removing the hydrogenated carbon atom and neglecting the structural reconstructions. Such descriptions preserve particle-hole symmetry and lead to the appearance of a quasi-bound state at the Dirac point. The zero-energy state is fully polarized on the sublattice opposite to that of the removed $p_z$ orbital and presents an algebraic decay due to the gapless relativistic spectrum[35–38].

In the spinless description, the zero-energy state exhibits a fractional charge $Q = -e/2$[16]. The quasi-bound state results from the promotion of half a state from the valence band and half a state from the conduction band. This is a two-dimensional analogue of the charge fractionalization at domain-wall solitons in the one-dimensional spinless model of polyacetylene[39,40], similar to that of the structural Kekulé vortex[5]. In contrast, the fractional excitations no longer exist in the spinfull description, which doubles the number of zero-energy states. At half-filling, one of the two spin-polarised states is fully occupied.

Since each bound state is a hybrid superposition of valence-conduction half-states of $-e/2$ and $+e/2$ fractional charges, the quasi-bound state around a vacancy must have a neutral charge $Q = 0$ with spin $S = 1/2$. Doping the sample then leads to a charged bound state $Q = \pm e$ with zero spin $S = 0$.

In realistic graphene systems with H adatoms, the particle-hole symmetry is not exactly respected. Thus, the bound state in real graphene is slightly shifted from zero energy and its charge should deviate from e/2. However, this deviation is very small. According to first-principle calculations[34], the shift is approximately $t/16$ ($t$ is the nearest-neighbour hopping), which is just 0.01 of the total $p_z$ bandwidth.

It is therefore particularly interesting that previous spectroscopic measurements apparently provide evidence of such spin-charge relations for the quasi-bound states associated with the Kekulé vortex texture in our sample[41]. In this earlier work, the state occupancy was tuned with the doping, and DOS measurements revealed the formation of quasi-bound $S = 1/2$ magnetic moments at half-filling and $S = 0$ in doped graphene. Furthermore, a small (see ref. 5 and Supplementary Information) shift of the zero energy bound state associated with particle-hole symmetry breaking could not be revealed[41]. These measurements are also consistent with previous DFT studies, in which the hydrogen chemisorption leads to charge neutral and $1\,\mu_B$ spin-polarized quasi-bound states[36]. These observations do not constitute a direct evidence of the charge fractionalization mechanism near H adatoms, but are sufficiently consistent with it to stimulate further experimental efforts to address the spin and charge state of the bound state as well as theoretical work to link the Berry phase with the charge fractionalization mechanism.

## Kekulé pattern near divacancies

The existence of the Kekulé vortex and the quasi-bound states induced by the hydrogen adatom defect is related to the symmetries of this scattering center. While a hydrogen adatom breaks the $C_2$ symmetry and sublattice balance of graphene, a divacancy preserves these symmetries. Figure 4a shows an STM image of such divacancy in graphene. The divacancy also induces locally a Kekulé pattern. Importantly, this Kekulé pattern is exempt of any winding, with the Kekulé domain being defined by one of the three orientations of the chemical bond linking the two removed atoms. This is confirmed by DFT calculations (Fig. 4c), which show that the defect also creates significant lattice distortion originating from structural reconstruction due of the formation of two pentagonal rings (Figs. 3c and 4d).

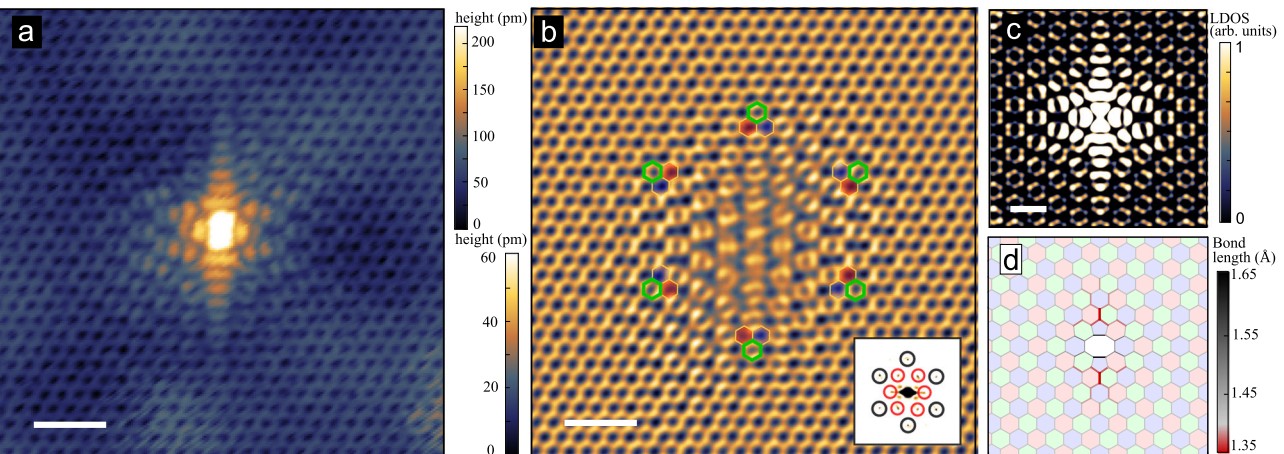

**Fig. 4 | Kekulé texture induced by the divacancy defect in graphene. a** An STM image of a divacancy in graphene ($V_b = 500$ mV, $i_t = 400$ pA). The scale bar is 1 nm. **b** Fourier filtered image to highlight the Kekulé bond texture. The Fourier filter is shown on the inset. Selected harmonics are those of graphene (black circles) and $\sqrt{3} \times \sqrt{3} R30°$ (red circles). **c** DFT simulated STM image. The scale bar is 1 nm. **d** Atomic structure of the divacancy defect in graphene from DFT calculations. The bond length were calculated from relaxed atomic positions.

A local Kékulé pattern with or without vortex can therefore be induced in graphene by the specific distribution of atomic defects. Harnessing experimentally these building blocks could lead to the long-awaited macroscopic Kekulé engineering from the cooperative effect of atomic defects[18,19].

## Methods

### Samples and STM measurements
The samples were grown by thermal decomposition of the carbon-face SiC at temperatures close to 1150°C in ultrahigh vacuum.[42] Silicon evaporation results in several graphene layers decoupled by rotational disorder. Hydrogen atoms were then deposited by thermal dissociation of hydrogen gas in a custom atomic hydrogen source as described previously.[41] STM images were obtained in the constant current mode in a custom ultrahigh vacuum setup at 5 K.

### Tight-binding calculations
The nearest-neighbor tight-binding Hamiltonian of graphene is expressed as

$$H = \sum_{\langle i,j \rangle} t c_i^\dagger c_j + h.c.$$
$$= t \begin{pmatrix} 0 & 1 + e^{ik_x} + e^{ik_y} \\ 1 + e^{-ik_x} + e^{-ik_y} & 0 \end{pmatrix}, \quad (3)$$

where the nearest-neighbor hopping integral $t = -2.7$ eV. The TB calculation is carried out with periodic boundary conditions, with the hydrogen adatom modelled as a large on-site potential ($V = 100|t| \approx 270$eV). The supercell size of $27 \times 27$ unit cells of graphene was used to reduce the spurious effects due to the mutual interference between the periodic images of adatoms.

The information about the phase difference between wavefunction amplitudes on the nearest-neighbor sites, that is the bond order, is needed in order to describe the Kekulé texture. We define the bond parameter as the LDOS between two neighboring atoms $i,j$. On the basis of TB model, we consider the wave function as the product of the TB eigenvector $\psi$ and an envelope function $f(r)\phi(r) = \sum_n \sum_i \phi_i^n f(r - r_i)$, taking the middle point between the atoms $r_{ij} = (r_i + r_j)/2$ LDOS writes

$$\rho(r_{ij}) = |f(a_0/2)(\psi_i + \psi_j)|^2$$
$$= f(a_0/2)(2Re\langle\psi_i|\psi_j\rangle + \psi_i^*\psi_i + \psi_j^*\psi_j) \quad (4)$$

in which the inner product $\langle\psi_i|\psi_j\rangle$ governs bond texture. Therefore, we use the orbital overlap $Re\langle\psi_i|\psi_j\rangle$ as the bond-order operators in the TB calculation. In the presence of a scattering center, $\langle\psi_i|\psi_j\rangle$ is perturbed by the inter-sublattice Green's functions $G_{AB}$ and $G_{BA}$ (See Supplementary Information), since the atoms connected by the bond are in different sublattices. The bond order is plotted by integrating the LDOS from 0 to 300 meV.

### DFT calculations
First-principles calculations were performed using the SIESTA code[43]. We use the double-$\zeta$ plus polarization localized orbital basis set combined with the local density approximation exchange-correlation functional[44]. The energy shift for constructing the localized orbital basis functions was set to 275 meV, and the real-space cutoff to 250 Ry. The structural relaxation was performed using the conjugate gradient method.

The simulated STM images were produced using the *plstm* module of the SIESTA package, as a postprocessing step following the DFT calculations. The images were simulated from LDOS at a constant height of 2.5 Bohrs above the graphene plane.

## Data availability
The data that support the findings of this study are available from the corresponding author upon request.

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

## Acknowledgements

V.T.R. acknowledges the support from the ANR Flatmoi project (ANR-21-CE30-0029). Y.G. and O.V.Y. acknowledge support from the Swiss National Science Foundation (grant No. 204254). Computations were performed at the Swiss National Supercomputing Centre (CSCS) under projects No. s1146 and the facilities of the Scientific IT and Application Support Center of EPFL. C.D. acknowledges support from the projects TED, CDS-QM, and TopoMat (ANR-23-CE30-0029), respectively funded by Quantum Matter Bordeaux, the SMR department of Bordeaux University, and the French Research National Agency. MMU acknowledges support by the European Research Council Consolidator Grant (No. 101087014) mKoire. I.B. acknowledges the support from the "(MAD2D-CM)-UAM" project funded by Comunidad de Madrid, by the Recovery, Transformation and Resilience Plan, and by NextGenerationEU from the European Union, the Spanish Ministry of Science and Innovation (Grant PID2020-115171GB-I00) and the Comunidad de Madrid NMAT2D-CM program under grant S2018/NMT-4511. H.G-H. acknowledges financial support from the Spanish State Research Agency under grant Ramón y Cajal fellowship RYC2021-031050-I

## Author contributions

H.G.-H., M.M.U. and I.B. performed the experiments under the supervision of I.B.. V.T.R. discovered the Kekulé vortex. Y.G. performed DFT and TB calculations under the supervision of O.V.Y. Y.G. and C.D. performed Green's function calculations. M.I.K. gave technical support and conceptual advice. Y.G., C.D., O.V.Y. and V.T.R. wrote the manuscript with the input of all authors. V.T.R. coordinated the collaboration.

## Competing interests

The authors declare no competing interests.
