## [Peer Review File · Nature Communications]

Observation of Kekulé vortices induced in graphene by hydrogen adatomsReviewers' Comments:

Reviewer #1:

Remarks to the Author:

This manuscript reports on the observation of a "Kekulé vortex" around a hydrogen adatom on graphene on SiC and the absence of such a vortex for a divacancy. I find the results of the authors' study convincing, scientifically interesting, and suitable for the audience of Nature Communications.

I have a few comments for improving the manuscript:

1. I suggest the authors tone down the discussion on fractionalized excitations in the abstract. The existence of fractionalized excitations in the presence of a hydrogen adatom is just conjecture, and the authors do not provide any experimental evidence.
2. The authors cite Mao et al. (Reference 38) for the charge neutrality of a hydrogen adatom, but that paper is for vacancies. Do the authors know if a monovacancy has a Kekulé vortex?
3. There are color wheels in Figs. 1a-d illustrating the continuous nature of the Kekulé order parameter, but how is this order parameter defined? Is it the angle ϕ in equation S5? Is the Kekulé order parameter actually changing at a uniform rate as one circles the hydrogen adatom, as illustrated?
4. Moreover, why is $|K'A\rangle$ and $|KB'\rangle$ missing in equation S5? This system is not the zeroth Landau level of graphene, although I guess the magnetic moment might break time reversal symmetry.
5. I would like to bring Nuckolls et al. (arXiv:2303.00024), which observes vortices in Kekulé patterns in magic-angle graphene, to the authors' attention. This paper constructs a continuous Kekulé order parameter (see my question #3) and also considers $|K'A\rangle$ and $|KB'\rangle$ (question #4).
6. The authors mention that the 2π winding of the Kekulé order parameter is reminiscent of the π Berry phase in graphene. If there is a connection between these ideas, could the authors elaborate?

Reviewer #2:

Remarks to the Author:

The authors report a discovery of the Kekule vortex in graphene with a dilute concentration of chemisorbed hydrogen atoms using scanning tunneling microscopy. Such Kekule vortices have been predicted theoretically to carry fractional electric charges and obey fractional exchange statistics (in idealized spinless case). Therefore, they are potentially interesting objects and their discovery merits publication in a high-profile journal.

The data presented in the manuscript is of high quality and it indeed supports the existence of Kekule vortices centered on chemisorbed hydrogen atoms. However, I do find the authors' discussion of the evidence very confusing. This must be improved before I can recommend acceptance. Below I list main points of confusion that I would like to see addressed:

1. The key Figure 1b and the surrounding discussion does not clearly explain what is it that the reader should be focusing their attention on. The colored overlays are not useful because they almost completely obscure the data. In my trying to understand the Kekule patterns I found most useful the presentation given in Extended Data Figure 2 panels (d) and (g). Here one clearly sees the variation of the LDOS relative to the fixed honeycomb lattice superimposed in such a way that it does not obscure the data. On the other hand the meaning of the honeycomb lattice has not been clearly explained. The main text on p. 2 says: "The pattern is dominated by the onsite signal on sublattice B (assuming the H adatom is on sublattice A), which alternates between the three nonequivalent B sites." This is clear and helpful. However, B sites form a triangular, not honeycomb, lattice. Would it then be more useful to overlay the relevant triangular lattice?
2. Statement on p. 2 "A closer inspection of the interference pattern further reveals the underlying signal on graphene bonds, which is consistent with the Kekule ordering" is cryptic and unhelpful. What

exactly is the reader supposed to inspect more closely?

3. The rationale for calling the vortex "Berry-Kekule" is not well explained. How exactly does the Kekule vortex derive from "an intrinsic topological property of the massless Dirac (that is, chiral in pseudospin) wavefunctions scattered by the adatom" ? Is this vortex different in some way from the Kekule vortex considered in Refs. [4,5] ? If not, then why is the new name necessary? If yes then how exactly?

4. Related to the above point (3) the authors do not seem to answer the question regarding the spectral gap generated by the vortex.

In summary, I find this to be a fundamentally sound and interesting experimental work which however suffers from poor presentation, as exemplified by the remarks above. An improved manuscript could be suitable for publication in Nature Communications.

Reviewer #3:

Remarks to the Author:

The authors present experimental evidence that a hydrogen adatom on graphene induces an hexagonal superlattice with a three times larger unit cell. State of the art STM images are presented which give compelling evidence that this adatom gives rise to a structure of the electronic density consistent with a Kekule texture centered at the location of the adatom.

This analysis is backed by extensive numerical DFT simulations providing further evidence for a Kekule structure.

The claim is very interesting, timely and suggestive. The experimental results are very neat and of the highest quality. The numerical analysis is interesting, which is also supported by a more microscopic calculation of the electronic density using the Green's function of graphene perturbed by an adatom.

For all these reasons, I recommend to publish this work in Nature Communications.

This being said, the claim of the authors is seriously debatable. In the original and subsequent works on Kekule distortions in graphene (Hou et al., Jackiw et al. 2007), it appeared that a Kekule order should preserve the chiral symmetry of pristine graphene. But it modifies graphene in a way which turns it into a non trivial topological phase. For a centro-symmetric field as considered by the authors, the Kekule order is essentially a vortex characterised by invariant integers and a fractional charge. Now the question is whether an adatom can create these conditions exactly or only approximately. The authors claim that an adatom is equivalent to a single atom vacancy. It seems that this claim is inaccurate despite undeniable similarities between the two problems in the limit of a very strong local potential (e.g. existence of zero modes). But an adatom does not share the topological content of a Kekule distortion or a vortex in the present case.

The input of the authors on these questions would be highly appreciated and undoubtedly helpful in interpreting the beautiful results presented in this work.

Dear Referees,

We first would like to thank the reviewers for their careful reading of our manuscript, their strong support of our work and for challenging our statements. This has allowed us to clarify the important issue of the relation of the Kekulé vortex winding to the Berry phase. This also lead us to realize that our previous spectroscopic results (González-Herrero Science **352**, 437 (2016)) are consistent with the unusual spin charge relation expected for the bound state. This simplifies the discussion as there is no need to invoke results on the cousin (yet different) experimental system of a vacancy. We perfectly agree that this is not a demonstration of fractional excitations and we made this explicit in the manuscript. Motivated by the referee's questions we made complementary DFT calculations on vacancies which evidence that they also induce a Kekulé vortex even if they are accompanied with a stronger structural relaxation as expected. This is now presented in the supplements. Finally, we have also profoundly restructured the manuscript to avoid repetitions of some of the content. All in all, the constructive suggestions of the referees helped us to strongly improve the manuscript (The main changes are highlighted in color in the manuscript). You will find below our response to their comments. Our replies appear in blue.

Sincerely yours,

The authors

REVIEWER COMMENTS

Reviewer N◦1 (Remarks to the Author):

This manuscript reports on the observation of a “Kekulé vortex” around a hydrogen adatom on graphene on SiC and the absence of such a vortex for a divacancy. I find the results of the authors' study convincing, scientifically interesting, and suitable for the audience of Nature Communications.

We would first like to thank the referee for their positive feedback on our work.

I have a few comments for improving the manuscript:

1. I suggest the authors tone down the discussion on fractionalized excitations in the abstract. The existence of fractionalized excitations in the presence of a hydrogen adatom is just conjecture, and the authors do not provide any experimental evidence.

We agree with the reviewer that we do not report direct experimental evidence of fractionalized excitations. In the abstract, we only motivate our study from the perspective of fractionalized excitations, which under time-reversal symmetry are anticipated *”as bound states of a rotating bond order known as Kekulé vor-*

tex. However, the physical mechanisms inducing such topological defects remain elusive, preventing experimental realizations. Here, we report the observation of Kekulé vortices in the local density of states of graphene under time-reversal symmetry. We believe that this abstract fairly motivates/announces the purpose of the manuscript: the experimental realisation of a Kekulé vortex. Here, we never claim to provide experimental evidence of fractionalized excitations, though we discuss it latter in the manuscript in connection to previous theoretical and experimental studies in the literature. Thus, we would like to keep the abstract as it is, though we shorten a bit the first sentence on the fractional quantum Hall effect.

That said, the feedback lead us to reconsider our results more globally. In the previous version of the manuscript, it was not clear that the Kekulé vortex discussed here is observed on the same H atom as discussed in the previous study (Fig. 2b of González-Herrero Science **352**, 437 (2016)). This H atom is chemisorbed on neutral graphene and we had demonstrated its magnetism by spectroscopic measurements, which show a split resonance at the Fermi energy. On the contrary, in the same study, we showed that H atoms deposited on doped graphene show a single resonance typical of a non-magnetic state. Therefore, even if this was not stated in this way, we have already demonstrated the spin 1/2 of the neutral state and spin 0 of the doped state consistent with the charge fractionalisation mechanism. We have changed the corresponding paragraphs and removed the reference to the work of Mao et al. dealing with carbon vacancies. This was leading to confusions even if the result for vacancies should be similar (see response to the next question).

2. The authors cite Mao et al. (Reference 38) for the charge neutrality of a hydrogen adatom, but that paper is for vacancies. Do the authors know if a monovacancy has a Kekulé vortex?

Motivated by the reviewer’s question, we have performed new DFT calculations on vacancies in graphene, both with and without atomic relaxation. These calculations show (see Figure 1 below) the existence of a Kekulé vortex in both cases. It should be noted that lattice reconstruction explicitly breaks the C_3 symmetry of vacancy sites in graphene. These results are presented in the revised supplementary material (see Figure S7 in the Supplementary Information and Figure 1 below).

3. There are color wheels in Figs. 1a-d illustrating the continuous nature of the Kekulé order parameter, but how is this order parameter defined? Is it the angle phi in equation S5? Is the Kekulé order parameter actually changing at a uniform rate as one circles the hydrogen adatom, as illustrated?

The angle defining the order parameter is the angle $\theta_{\mathbf{r}}$ involved in the bond contribution to the LDOS in Eq. (1). It corresponds to the real-space representation of the pseudospin and varies at a uniform rate around the H adatom. We

Figure 1: Top row: Structure and DFT simulated STM image of a vacancy in graphene without relaxation. Bottom row: Structure and DFT simulated STM image of a vacancy in graphene with relaxation. C_3 symmetry is broken by the bond reconstruction but a clear Kekulé vortex is seen far from the vacancy.

thank the reviewer for this remark. We now define it in the legend of Fig. 1 in the revised version of the manuscript.

4. Moreover, why is $|K'A\rangle$ and $|KB'\rangle$ missing in equation S5? This system is not the zeroth Landau level of graphene, although I guess the magnetic moment might break time-reversal symmetry.

Equation S5 intends to provide a definition of Kekule order through intervalley coherence. The STM results probe the LDOS on **bonds** among other contributions, which must be represented by the overlap of wavefunction amplitudes on A and B sublattice atoms. As shown in Fig. S2, the intervalley coherence guarantees that the periodicity of bond order is the same as the Kekule lattice ($L_{kek} = \sqrt{3}a$). Furthermore, phase ϕ determines the flavor of the Kekulé pattern, i.e. one of the Kekulé sublattices characterized by stronger bonds. The time-reversal counterpart will provide the same amplitude of modulation on the bonds, while we only write the $K - K'$ part to explicitly demonstrate the effect of ϕ .

The vortex order comes from the scattering on the point defect. On the bond, the scattering process involves incoming electrons from site A (resp. B) that scatter on the adatom and go back to the STM tip through site B (resp. A). We also focus on intervalley scattering: intravalley scattering does not provide any vortex winding. Thus, the initial and final states on the bond LDOS results from scattering processes of the form $\langle K, A | K', B \rangle$.

5. I would like to bring Nuckolls et al. (arXiv:2303.00024), which observes vortices in Kekulé patterns in magic-angle graphene, to the authors attention. This paper constructs a continuous Kekulé order parameter (see my question number 3) and also considers $|K'A\rangle$ and $|KB'\rangle$ (question number 4).

We thank the reviewer for bringing this beautiful work on MATBG to our attention. However this result is not directly related to our study (TBL vs monolayer graphene) and we prefer not to cite it.

6. The authors mention that the 2π winding of the Kekulé order parameter is reminiscent of the π Berry phase in graphene. If there is a connection between these ideas, could the authors elaborate?

The Kekulé vortex in the LDOS resolved at the STM tip comes from the bond contribution in Eq.(1). The vortex refers to the angle $\theta_{\mathbf{r}}$, which is singular at the origin. This angle indexes the STM tip orientation around the H atom. In a quasi-particle interference (QPI) image, where the signal is dominated by backscattering, this angle relates to the wavevector orientation $\theta_{\mathbf{q}}$ of the incoming wavefunction on the defect (see inset of Fig. 1b and discussions in Dutreix et al., Nature (2019)). The wavevector orientation $\theta_{\mathbf{q}}$ is defined with respect to a Dirac point in momentum space. In graphene, the momentum is locked to the sublattice pseudospin, whose winding around a Dirac cone is

twice the Berry phase. Therefore, by cycling the STM tip around the adatom, one reveals the Kekulé vortex charge, which directly roots down the pseudospin winding, and so the Berry phase. Following the reviewer's remark, we have made more explicit this connection to the Berry phase through the lock-in relation between the tip orientation θ_r and the sublattice pseudospin θ_q .

The paragraph now reads

The vortex originates from the polar angle θ_r indexing the tip orientation. We have previously established that in such quasi-particle interference signal which is dominated by back-scattering, this angle is a real-space representation of the electron pseudospin θ_q defined from a Dirac point in momentum space (see inset in Fig. 1b and Ref. 25). Thus, the bond contribution to the LDOS exhibits a 2π vortex centered on the adatom as a reminiscence of the topological pseudospin vortex, or equivalently the topological Berry phase, which characterizes the band topology of the massless Dirac electrons.

Reviewer N°2 (Remarks to the Author):

The authors report a discovery of the Kekule vortex in graphene with a dilute concentration of chemisorbed hydrogen atoms using scanning tunneling microscopy. Such Kekule vortices have been predicted theoretically to carry fractional electric charges and obey fractional exchange statistics (in idealized spinless case). Therefore, they are potentially interesting objects and their discovery merits publication in a high-profile journal.

The data presented in the manuscript is of high quality and it indeed supports the existence of Kekule vortices centered on chemisorbed hydrogen atoms. However, I do find the authors' discussion of the evidence very confusing. This must be improved before I can recommend acceptance. Below I list main points of confusion that I would like to see addressed:

We thank the referee for their support and comments on our work.

1. The key Figure 1b and the surrounding discussion does not clearly explain what is it that the reader should be focusing their attention on. The colored overlays are not useful because they almost completely obscure the data.

In my trying to understand the Kekule patterns I found most useful the presentation given in Extended Data Figure 2 panels (d) and (g). Here one clearly sees the variation of the LDOS relative to the fixed honeycomb lattice superimposed in such a way that it does not obscure the data. On the other hand the meaning of the honeycomb lattice has not been clearly explained. The main text on p. 2 says: "The pattern is dominated by the onsite signal on sublattice B (assuming the H adatom is on sublattice A), which alternates between the three nonequivalent B sites." This is clear and helpful. However, B sites form a triangular, not honeycomb, lattice. Would it then be more useful to overlay the relevant triangular lattice?

It is indeed possible to use the Kekulé lattice (that is the $\sqrt{3} \times \sqrt{3}R30^\circ$

lattice) as suggested by the referee and done in Figs. S2d and S2g in the Supplementary Information. This is quite useful when looking at the onsite (meaning on carbon atoms) signal, which was the subject of our previous publication (Dutreix et al Nature (2019)). However, we do not find it satisfactory to discuss the Kekulé signal of interest for the present study, which lies on the carbon-carbon bonds and requires the support of the atomic lattice to clearly see the stronger-bonds donuts-like patterns as sketched in Fig. 1a. This is why we have kept the traditional way of representing the Kekulé lattice with rgb color tiles in Fig. 1b. In this figure, the on-site (once again meaning on-carbon-atom) signal is highlighted by coloured disks. The color is defined by which carbon atom in 'ON' within the $\sqrt{3} \times \sqrt{3}R30^\circ$ lattice. Following a path encircling the H atom, one can see that this signal follows the r,g,b,r,g,b sequence demonstrating the 4π vortex we have discussed previously.

The Kekulé signal of interest for the present paper lies on carbon-carbon bonds (between carbon atoms) and is of a weaker intensity. In Fig. 1b, the donuts-like Kekulé signal is overlaid with coloured hexagons to highlight it. We do not agree with the referee that this hides the signal. One should look right next to the overlay to see the Kekulé signal and convince themselves that it is well in phase with the overlay. Winding around the H atom, this signal follows the rgb sequence, hence the 2π vortex.

We believe that Fig. 1b is the best we can do to summarize the complex and diverse information available from the STM image and we have modified the corresponding part in the manuscript to make the clearest statement on what the reader should look at.

2. Statement on p. 2 "A closer inspection of the interference pattern further reveals the underlying signal on graphene bonds, which is consistent with the Kekule ordering" is cryptic and unhelpful. What exactly is the reader supposed to inspect more closely?

We thank the reviewer for this remark. We have rephrased the discussion as follows: "*A closer inspection of the interference pattern also reveals a strong donut-like signal on some carbon-carbon bonds, which we highlight with bold coloured hexagons in Fig. 1b. This color tiling enables us to identify three distinct Kekulé domains separated from one another by domain walls along the armchair directions. Remarkably, the Kekulé bond order shows a 2π winding when circling the STM tip around the adatom. This demonstrates that the chemisorbed H adatom induces a Kekulé vortex on the surrounding graphene bonds*". We hope that this is clearer and more helpful.

3. The rationale for calling the vortex "Berry-Kekule" is not well explained. How exactly does the Kekule vortex derive from "an intrinsic topological property of the massless Dirac (that is, chiral in pseudospin) wavefunctions scattered by the adatom" ? Is this vortex different in some way from the Kekule vortex considered in Refs. [4,5] ? If not, then why is the new name necessary? If yes then how exactly?

The Kekulé vortex we observe is indeed different from the one in Refs. [4,5].

First, the vortex in the Ref. [4] results from *bond distortions*. It is introduced as an ad hoc assumption (i.e. without specifying the microscopic mechanism responsible for it) and implies a change of graphene atomic structure. In contrast, we propose an explicit microscopic mechanism – elastic scattering of the electrons on a point defect – and it does not necessarily involve any bond distortions. We explain very well our observations using tight-binding, low-energy Green’s function, and DFT approaches that do not account for bond distortions. We also present DFT calculations that include the atomic relaxation. These calculations confirm that the bond distortions are negligible at the H adatom.

Second, the vortex in Ref. [4] is introduced through an ad hoc potential in the Hamiltonian. In our study, the adatom potential is real and does not show any vortex. The vortex – i.e. the phase singularity – originates from an intrinsic property of the massless relativistic electrons scattering on the adatom: the pseudospin phase $e^{i\theta_{\mathbf{q}}}$ between the sublattice components of the spinor wavefunction $|\psi\rangle \propto |A\rangle + e^{i\xi\theta_{\mathbf{q}}}|B\rangle$. The pseudospin winding relates to the Berry phase gained by the wavefunction cycling around a Dirac point. Since the QPI signal of interest in our STM image is dominated by back-scattering and the pseudospin is locked on the momentum via $\theta_{\mathbf{q}}$, and since $\theta_{\mathbf{q}}$ is locked on the STM tip orientation $\theta_{\mathbf{r}}$ (see inset in Fig 1b), cycling the STM tip around the adatom is like cycling the pseudospin around the Dirac point in momentum space. Thus, the charge of the Kekulé vortex is directly determined by the pseudospin winding, and so the Berry phase, around a Dirac point.

We hope that this discussion clarifies the naming Berry-Kekulé and why the vortex we observe is different from that in Refs. [4,5]. Following the reviewer’s remark, we have made the connection to the Berry phase more explicit in the revised version of the manuscript.

4. Related to the above point (3) the authors do not seem to answer the question regarding the spectral gap generated by the vortex.

We thank the reviewer for this remark. In addition to the differences in origin in point (3), a H adatom chemisorbed in graphene does not open a gap. While the Kekulé-vortex bound state in Refs. [4,5] lies within the Kekulé band gap and is exponentially localized, the Berry-Kekulé vortex bound state couples to the continuum and is quasi-localised with an algebraic decay (see e.g. Refs. [29-31]). We have added this discussion to the revised version of the manuscript. The following sentence specifies this in the main text.

The zero-energy state is fully polarized on the sublattice opposite to that of the removed pz orbital and presents an algebraic decay due to the gapless relativistic spectrum.

In summary, I find this to be a fundamentally sound and interesting exper-

imental work which however suffers from poor presentation, as exemplified by the remarks above. An improved manuscript could be suitable for publication in Nature Communications.

We thank the reviewer for their comment and hope that the clarifications above make the manuscript suitable for publication in Nature Communications.

Reviewer N°3 (Remarks to the Author):

The authors present experimental evidence that a hydrogen adatom on graphene induces an hexagonal superlattice with a three times larger unit cell. State of the art STM images are presented which give compelling evidence that this adatom gives rise to a structure of the electronic density consistent with a Kekule texture centered at the location of the adatom.

This analysis is backed by extensive numerical DFT simulations providing further evidence for a Kekule structure.

The claim is very interesting, timely and suggestive. The experimental results are very neat and of the highest quality. The numerical analysis is interesting, which is also supported by a more microscopic calculation of the electronic density using the Green's function of graphene perturbed by an adatom.

For all these reasons, I recommend to publish this work in Nature Communications.

This being said, the claim of the authors is seriously debatable. In the original and subsequent works on Kekule distortions in graphene (Hou et al., Jackiw et al. 2007), it appeared that a Kekule order should preserve the chiral symmetry of pristine graphene. But it modifies graphene in a way which turns it into a non trivial topological phase. For a centro-symmetric field as considered by the authors, the Kekule order is essentially a vortex characterised by invariant integers and a fractional charge. Now the question is whether an adatom can create these conditions exactly or only approximately. The authors claim that an adatom is equivalent to a single atom vacancy. It seems that this claim is inaccurate despite undeniable similarities between the two problems in the limit of a very strong local potential (e.g. existence of zero modes). But an adatom does not share the topological content of a Kekule distortion or a vortex in the present case. The input of the authors on these questions would be highly appreciated and undoubtedly helpful in interpreting the beautiful results presented in this work.

We thank the reviewer for their strong endorsement of our work and this very insightful remark.

First, we would like to explain the origin of Kekuleé order in these works. In Hou et al 2007, the Kekule order is introduced as an **intrinsic** bond order in the system, while our work is focusing on the **emergent** bond order triggered by the adatom. We further compare the zero mode at Kekule vortex in [Hou

2017] and the analytical solution of vacancy zero modes, which may suggest that there is a fractionalization in the single-vacancy or adatom case.

In the first-order approximation, both a single vacancy and the adatom contribute to the LDOS as an on-site scatterer that breaks sublattice symmetry. Therefore, we treat them similarly in the study of bond order and both of them are expected to induce a bond order vortex. Nevertheless, we agree with the Reviewer that the adatom is quantitatively different from a vacancy. The difference lies in the chiral-symmetry breaking effects of the two kinds of defects. In our work, however, the vortex in bond order originates from the Berry phase of graphene as such, and not from the topology of the combined bulk-defect system.

Our new DFT result shows that both a vacancy and an H adatom induce a Kekulé vortex in their surrounding. This should be verified experimentally for the vacancy, while we expect that the adatom is slightly more favorable since the vacancy leads to a bond reconstruction that breaks the C_3 symmetry.

Reviewers' Comments:

Reviewer #1:

Remarks to the Author:

I understand fractional excitations are a motivation for the study, but I still feel the beginning of the abstract puts too much weight on the concept, especially given the paucity of the evidence for it. As far as I am aware, there is no evidence that hydrogen adatoms on graphene have any charge, nor are there spin-resolved measurements (although, as the authors point out, González-Herrero et al. (Science 2016) infers the spin state through point spectroscopy). But ultimately, this is a stylistic concern, and the authors clearly do not wish to modify their abstract.

I find the authors' reply to my question about the color wheel unsatisfactory. Figure 1 is difficult to interpret, which is evident also from Referee #2's comments. The color wheels are clearly not data, and are purely schematic. Then, are we to just trust the authors that the intervalley interference phase angle tracks the polar angle of the STM tip location without any quantitative analysis? What if the phase winds almost to 2π and then winds back to zero as you circle around the adatom? This is why I pointed the authors to Nuckolls et al., which observes a Kekulé vortex by directly extracting order parameters from data. The authors should do something similar or present their own quantitative analysis method.

Why do the authors believe the time-reversed counterpart will have the same amplitude of modulation? Doesn't the adatom have magnetism and thus breaks time-reversal symmetry?

The relationship between a Kekulé vortex and the Berry phase of graphene is still unclear to me. The authors claim that (1) the tip location polar angle is equivalent to the wavevector orientation and (2) the intervalley interference phase is equivalent to the pseudospin angle (which is the A/B sublattice degree of freedom). But since valley and sublattice are not identical here (unlike in quantum Hall), neither of these are obvious to me.

I do, however, appreciate the authors' new discussion on Clar's sextets.

Reviewer #2:

Remarks to the Author:

I read the revised (and improved) version of the manuscript as well as the author response. Various cryptic remarks have been removed and the results are now presented with more clarity. On this basis I am happy to recommend acceptance, provided the authors address one remaining point listed below.

My comment relates to point (3) raised in my original report. I thank the authors for adding a detailed explanation for their notion of "Berry-Kekule" vortex -- as a result I now understand their rationale for this designation. At the same time I must question the necessity of using the new name.

Fundamentally, this is still the same Kekule vortex introduced by Hou et al. in Ref. [5]. Although Hou et al. did not emphasize the Berry phase aspect as done in the revised manuscript, the physics was necessarily there. The present authors emphasize the purely electronic origin of the vortex but mention a small lattice distortion found in their DFT calculation. Indeed as a general rule any electronic symmetry breaking must be accompanied by a lattice distortion of the same symmetry. So any difference between the present vortex and that of Hou et al. is of quantitative, not qualitative nature. For this reason I believe the object experimentally discovered in the present manuscript does not require a new name. To minimize confusion and improve accuracy the authors should stick to the original name "Kekule vortex".

Reviewer #3:

Remarks to the Author:

Second review on the manuscript : Observation of Kekule vortices induced in graphene by hydrogen adatoms.

In its present form and despite my unchanged positive appreciation for the high quality experimental results, I do not recommend the publication of this new version in Nature Comm.

I wish now to elaborate.

In my previous report, I indicated that "the claim of the authors, namely the observation of a Kekule vortex, is seriously debatable".

I challenged that claim and stated that a single adatom cannot create the conditions needed to observe a topological vortex. Reading the response of the authors to the reviewers, I am even more convinced that the statements of the authors and the presentation and interpretation of their results is misleading and inaccurate.

A main issue is about the exact meaning or understanding of a topological excitation, a Kekule vortex in the present case. In long and rather unclear elaborations, the authors eventually propose a new name for their topological excitation : the Berry-Kekule vortex even different from their title (Observation of Kekule vortices...).

Let us try to present the problem at stake:

1. Prediction and observation of (intrinsic or created) topological excitations with fractional charges constitute a "Holy Grail" in the field especially in the presence of time reversal symmetry.
2. Among different predictions, those associated to creation of topological defects are particularly interesting, since starting from materials void of topological properties (e.g. pristine graphene), fractional charge topological excitations can be obtained through a careful engineering of such defects.
3. A particularly interesting proposal presented in Ref.5, made use of a modulated tight-binding hopping akin to a Kekule texture. As emphasized by the authors in their reply, this Kekule texture is intrinsic and one of its essential property is that it preserves time reversal, particle-hole hence chiral symmetries. In the classification of possible topological orders (see e.g. Ref. 4), this Kekule texture is characterised by an integer winding number closely related to the existence of a fractional charge and of (protected) zero energy states.

Based on these features, the authors claim that such an "emergent Kekule bond order can be triggered from the existence of a hydrogen adatom in graphene". Their claim is that this emergent Kekule order shares some if not all of the topological features mentioned in the previous points. Following remarks made by the reviewers, these claims are toned down and nuances are indeed brought in, e.g by means of an abundant use of "reminiscent of". Yet, the authors in the added discussion in page 4 (and elsewhere) do not hide their claim that their Berry-Kekule vortex has many of the attributes e.g. fractional charge and zero modes as for structural Kekule vortex.

It is the opinion of this reviewer that these interpretations are misleading and inaccurate and a Kekule vortex with topological attributes cannot be emulated using a single adatom. First, a single adatom breaks particle-hole symmetry hence chiral symmetry so that in a two-dimensional system, it cannot correspond to any kind of topological order (see e.g. Ref.4). This is to be contrasted with the case of a vacancy which preserves chiral symmetry. The authors do not write the Hamiltonian they use in their calculations (see SM part III), yet it seems to be the same Hamiltonian used in Ref. 23 by some of the co-authors of the present paper, a Hamiltonian which clearly breaks chiral symmetry (and particle-hole). As a result of particle-hole symmetry breaking, Friedel oscillations show up as thoroughly

studied again in Ref.23. These oscillations impede the formation of a topological order (unlike a vacancy), and they prevent the existence of a true (i.e. topologically protected) zero mode.

DFT based STM images displayed in Fig.2.b, present a relaxed atomic structure of graphene with an adatom. While being neat and convincing, these images must be consistent with experimental STM images presented in part I of the SM, which are identical to STM images in Fig.1.a of Ref.23 obtained in the very same setup which displays Friedel oscillations in a relaxed atomic structure.

It seems therefore that there is a serious contradiction between raw STM images, numerics and calculations presented in parts I and II of the SM on one side and Fig.2.b in the main text based on DFT on the other side.

Moreover, the existence of an induced Kekule vortex is in contradiction with the breaking of particle-hole symmetry, a symmetry needed to observe any type of topological order in 2d in the presence of time reversal symmetry (according to the tenfold classification of topological insulators and superconductors, see e.g Ref.4).

To give a firmer basis to their claim, the authors must answer these points and discuss in more details the atomic relaxation process which, they claim, seems to play a role.

Reply to Reviewer 1

I understand fractional excitations are a motivation for the study, but I still feel the beginning of the abstract puts too much weight on the concept, especially given the paucity of the evidence for it. As far as I am aware, there is no evidence that hydrogen adatoms on graphene have any charge, nor are there spin-resolved measurements (although, as the authors point out, González-Herrero *et al.* (Science 2016) infers the spin state through point spectroscopy). But ultimately, this is a stylistic concern, and the authors clearly do not wish to modify their abstract.

I find the authors' reply to my question about the color wheel unsatisfactory. Figure 1 is difficult to interpret, which is evident also from Reviewer 2's comments. The color wheels are clearly not data, and are purely schematic. Then, are we to just trust the authors that the intervalley interference phase angle tracks the polar angle of the STM tip location without any quantitative analysis? What if the phase winds almost to 2π and then winds back to zero as you circle around the adatom? This is why I pointed the authors to Nuckolls *et al.*, which observes a Kekulé vortex by directly extracting order parameters from data. The authors should do something similar or present their own quantitative analysis method.

We are very grateful to the Reviewer for clarifying their initial remark. Following their suggestion, we now show direct evidence of the Kekulé vortex from the raw data, in the spirit of the work of Nuckolls *et al.* The key is that the signals on sublattice A, sublattice B, and on the bonds connecting the two, each transform under different irreducible representations of C_3 at the intervalley scattering wave-vectors in Fourier space. This allows us to extract the Kekulé order parameter, i.e bond-signal intensity and phase using a modified Geometric Phase Analysis (Hitch *et al.* Ultramicroscopy **74**, 131–146(1998)). We show the real-space extractions of the Kekulé order parameters for sublattice A, B, and the bonds in Fig.1 below. The bond order parameter exhibits a 2π phase winding centered on the H adatom, thus demonstrating the existence of the Kekulé vortex. The procedure assessing the order parameters is detailed in the SI.

FIG. 1. **Evaluation of the Kekulé order parameter.** **a**, An STM image of a H adatom on graphene (same as Fig. 1b of the main text). **b,c** Extracted Kekulé signal and its phase. **d,e** Extracted signal on the B sublattice and its phase. **f,g** Extracted signal on the A sublattice and its phase. Images in panels **a-f** are all $7.2 \text{ nm} \times 7.2 \text{ nm}$ in size. In **c**, **e** and **f** the amplitude is overlaid as a guide for the eyes. Also, a white area symbolizes the central region where the Kekulé signal cannot be extracted because of the strong signal from the H adatom.

Why do the authors believe the time-reversed counterpart will have the same amplitude of modulation? Doesn't the adatom have magnetism and thus breaks time-reversal symmetry?

We thank the Reviewer for this comment. The magnetic moment of the bound states appears only at half filling, below 25 meV, due to electron interactions. However, we probe the system at a larger energy, where the density of states of graphene has a single-electron character. Thus, we can safely disregard magnetism in our analysis. Besides, our experimental data is very well reproduced by single-electron theoretical models. In addition, we provide the DFT simulations in the spin-polarized case (Fig. 2), which exhibits the same behavior as predicted by the other theoretical models.

FIG. 2. Spin-polarized DFT simulation for the STM images. The total LDOS is integrated between 0 and 300 meV. **a**, Real-space image. **b-c** the A-sublattice signal for the $\sqrt{3} \times \sqrt{3}R30^\circ$. **d-e** the bond signal. **f-g** The signal on the B-sublattice.

The relationship between a Kekulé vortex and the Berry phase of graphene is still unclear to me. The authors claim that (1) the tip location polar angle is equivalent to the wavevector orientation and (2) the intervalley interference phase is equivalent to the pseudospin angle (which is the A/B sublattice degree of freedom). But since valley and sublattice are not identical here (unlike in quantum Hall), neither of these are obvious to me.

We thank the Reviewer for rephrasing their previous remark, which we clarify point-by-point below.

Point (1) is a general scattering argument. The fluctuations of local density of states resolved by STM consist of standing-wave interferences, which are dominated by back-scattering. This implies an incoming wave from the tip to the adatom and an outgoing wave backscattered from the adatom to the tip. Therefore, the polar angle $\theta(\mathbf{r})$ of the tip and that of the wave vector of the outgoing wave $\theta(\mathbf{q}_{\text{out}})$ verify $\theta(\mathbf{r}) = \theta(\mathbf{q}_{\text{out}})$, where we define both angles with respect to the x axis. Thus, the orientations of the STM tip and the outgoing wave-vector with respect to the adatom are locked onto one another (see Fig.3 below).

FIG. 3. Diagrammatic illustration of the relation between STM tip position $\theta(\mathbf{r})$ and $\theta(\mathbf{q}_{\text{out}})$. Within the plane-wave regime, the backscattering wave has the polar angle $\theta(\mathbf{q}_{\text{out}}) = \theta(\mathbf{r})$.

Point (2) implies the existence of sublattice pseudospin in graphene. This is characterized by the relative phase between the sublattice components of the spinor wave-function $\sqrt{2}|\psi\rangle = |A\rangle + e^{i\theta(\mathbf{q})}|B\rangle$ within a given valley. The winding W of the pseudospin along a closed path \mathcal{C} relates to the Berry phase

$$\varphi_B = i \oint_{\mathcal{C}} \langle \psi | \nabla_{\mathbf{q}} \psi \rangle \cdot d\mathbf{q} = \frac{1}{2} \oint_{\mathcal{C}} d\theta = \frac{W}{2}. \quad (1)$$

Unlike the quantum Hall effect, the electron wavefunctions are not valley-sublattice polarized in our scattering problem. However, the adatom is populating a particular sublattice, i.e. it is sublattice polarized. Let us consider

an adatom chemisorbed on sublattice A. Then, the relative phase between the sublattice components of the spinor wavefunction is necessarily involved for waves propagating between any B site and the A site of the adatom. This is the reason why it appears in our expression of the LDOS fluctuations. Given the lock-in relation in point (1), resolving the signal around the adatom reveals the pseudospin winding or, equivalently here, the topological Berry phase. This is the very same argument as used in Ref.[23].

To conclude, we would like to thank the Reviewer for helping us in improving the clarity of our manuscript.

Reply to Reviewer 2

I read the revised (and improved) version of the manuscript as well as the author response. Various cryptic remarks have been removed and the results are now presented with more clarity. On this basis I am happy to recommend acceptance, provided the authors address one remaining point listed below.

My comment relates to point (3) raised in my original report. I thank the authors for adding a detailed explanation for their notion of "Berry-Kekule" vortex – as a result I now understand their rationale for this designation. At the same time I must question the necessity of using the new name. Fundamentally, this is still the same Kekule vortex introduced by Hou et al. in Ref. [5]. Although Hou et al. did not emphasize the Berry phase aspect as done in the revised manuscript, the physics was necessarily there. The present authors emphasize the purely electronic origin of the vortex but mention a small lattice distortion found in their DFT calculation. Indeed as a general rule any electronic symmetry breaking must be accompanied by a lattice distortion of the same symmetry. So any difference between the present vortex and that of Hou et al. is of quantitative, not qualitative nature. For this reason I believe the object experimentally discovered in the present manuscript does not require a new name. To minimize confusion and improve accuracy the authors should stick to the original name "Kekule vortex".

We are pleased to read that our last explanation on the Berry phase origin of the vortex has been useful. Our intention to use the new name *Berry-Kekulé vortex* was to emphasize this origin. After reading the Reviewers' reports, we realise that it may have brought some confusion. We then follow the Reviewer's suggestion and now use the original name "*Kekulé vortex*" throughout the manuscript. We thank the Reviewer for addressing this point and helping us to improve the manuscript.

Reply to Reviewer 3

In its present form and despite my unchanged positive appreciation for the high quality experimental results, I do not recommend the publication of this new version in Nature Comm. I wish now to elaborate.

In my previous report, I indicated that "the claim of the authors, namely the observation of a Kekule vortex, is seriously debatable". I challenged that claim and stated that a single adatom cannot create the conditions needed to observe a topological vortex. Reading the response of the authors to the Reviewers, I am even more convinced that the statements of the authors and the presentation and interpretation of their results is misleading and inaccurate.

A main issue is about the exact meaning or understanding of a topological excitation, a Kekule vortex in the present case. In long and rather unclear elaborations, the authors eventually propose a new name for their topological excitation : the Berry-Kekule vortex even different from their title (Observation of Kekule vortices...).

We thank the Reviewer for pointing out this omission in the title. As mentioned to Reviewer 2 above, our intention to use the new name *Berry-Kekulé vortex* was to emphasise the Berry phase π (or topological pseudospin winding) as key origin of the bond vortex we observe in the Kekulé modulation. We now realise that it may have brought some confusion. We then follow the suggestion of Reviewer 2 and simply use the original name *Kekulé vortex*.

Let us try to present the problem at stake:

1. Prediction and observation of (intrinsic or created) topological excitations with fractional charges constitute a "Holy Grail" in the field especially in the presence of time reversal symmetry.
2. Among different predictions, those associated to creation of topological defects are particularly interesting, since starting from materials void of topological properties (e.g. pristine graphene), fractional charge topological excitations can be obtained through a careful engineering of such defects.
3. A particularly interesting proposal presented in Ref.5, made use of a modulated tight-binding hopping akin to a Kekule texture. As emphasized by the authors in their reply, this Kekule texture is intrinsic and one of its essential property is that it preserves time reversal, particle-hole hence chiral symmetries. In the classification of possible topological orders (see e.g. Ref. 4), this Kekule texture is characterised by an integer winding number closely related to the existence of a fractional charge and of (protected) zero energy states.

Based on these features, the authors claim that such an "emergent Kekule bond order can be triggered from the existence of a hydrogen adatom in graphene". Their claim is that this emergent Kekule order shares some if not all of the topological features mentioned in the previous points. Following remarks made by the Reviewers, these claims are toned down and nuances are indeed brought in, e.g by means of an abundant use of "reminiscent of". Yet, the authors in the added discussion in page 4 (and elsewhere) do not hide their claim that their Berry-Kekule vortex has many of the attributes e.g. fractional charge and zero modes as for structural Kekule vortex.

We suspect there might be some misunderstanding with regard to our claim, which we would like to clarify.

1. Our claim is the observation of a Kekulé bond texture with a 2π -winding vortex. This is the electronic structure that we call Kekulé vortex. We are very confident in our observation and its interpretation. First, the observation is very well reproduced by 4 different theoretical descriptions — low-energy Green's function analytics, tight-binding model calculations, DFT simulations, and Clar's sextet theory. As far as we understood, this was also acknowledged by the Reviewer in their first report: "*State of the art STM images are presented which give compelling evidence that this adatom gives rise to a structure of the electronic density consistent with a Kekule texture centered at the location of the adatom. This analysis is backed by extensive numerical DFT simulations providing further evidence for a Kekule structure. The claim is very interesting, timely and suggestive. The experimental results are very neat and of the highest quality. The numerical analysis is interesting, which is also supported by a more microscopic calculation of the electronic density using the Green's function of graphene perturbed by an adatom*".

Second, our claim was further reinforced following the suggestion of Reviewer 1 to separate the contributions of the two sublattices and bonds in the STM signal. This allowed us to extract directly the intensity and phase of the bond contribution to the signal. The result is shown in Figs. 1b,c in the reply to Reviewer 1. While the intensity reveals the Kekulé modulations, the phase clearly exhibits a 2π winding around the H adatom. This new analysis of the data establishes firmly the existence of the Kekulé vortex. It is also in agreement with our previous theoretical descriptions.

Thus, we are surprised to read the Reviewer’s statement that “*a single adatom cannot create the conditions needed to observe a topological vortex*”.

2. We do not claim that the bound states in our observation of the Kekulé vortex have fractional charges. We acknowledged this explicitly in our first reply to the Reviewers. In the main text, we simply discuss the possibility for the bound states to have fractional charges as an open question. Indeed, anticipations of Kekulé vortex in graphene are associated with fractional excitations (main text ref. [5]). In order to avoid any ambiguity we have reformulated the end of our discussion on the spin-charge relation of the bound states with the following sentence:

It is therefore particularly intriguing that previous spectroscopic measurements apparently provide evidence of such spin-charge relations for the quasi-bound states associated to the Kekulé vortex texture in our sample[1]. In this earlier work, the state occupancy was tuned with the doping and DOS measurements reveal the formation of quasi-bound magnetic moments $S = 1/2$ at half-filling and $S = 0$ for doped graphene. Furthermore, the small (see Ref. 2 and Supplementary Information) shift of the zero energy bound state associated to particle-hole symmetry breaking could not be revealed[1]. These measurements are also consistent with previous DFT studies, in which the hydrogen chemisorption leads to charge neutral and $1\mu_B$ spin-polarized quasi-bound states[3]. These observations do not constitute direct evidence of the charge fractionalization mechanism near an H atom, but are sufficiently consistent with it to justify further experimental efforts for a direct measure of the spin and charge states of the bound state and theoretical work to link the Berry phase to the charge fractionalisation mechanism.

First, a single adatom breaks particle-hole symmetry hence chiral symmetry so that in a two-dimensional system, it cannot correspond to any kind of topological order (see e.g. Ref.4). This is to be contrasted with the case of a vacancy which preserves chiral symmetry.

We agree with the statements of the Reviewer regarding the relation between symmetries and topological order. However, we suspect there is fair degree of misunderstanding with regard to the actual structure and effects of H adatoms and vacancies in graphene. Let us elaborate.

Firstly, a simple model of single-atom vacancy defect in graphene consists in removing one site from the honeycomb lattice together with the nearest-neighbor hopping amplitudes. We understand that such a model does not break chiral symmetry and belongs to the BDI class with topological order. We stress, however, that this model of vacancy in graphene is oversimplified as it neglects a local structural reconstruction that is widely observed in experiments [see e.g. Fig. 3f in J. C. Meyer *et al.* Nano Lett. **8**, 3582–3586 (2008)]. Such a reconstruction results in the formation of an extra covalent bond between two undercoordinated carbon atoms. This bond couples two atoms within the same sublattice, and hence leads to a strong breaking of particle-hole symmetry. Thus, a real vacancy clearly breaks chiral symmetry. Strictly speaking, realistic graphene with a single vacancy no longer exhibits the BDI-class topological order.

Hydrogen adatoms are similar to vacancies in a sense of having an effect of removal of one site from the lattice. Hydrogen forms a covalent bond to one of the carbon atoms resulting in its rehybridization to the sp^3 hybridization state, that is “disabling” its p_z atomic orbital, and hence effectively acting as a vacancy. In contrast to the real vacancy, there is no reconstruction leading to the direct coupling of atoms in the same sublattice. Only a weak, residual particle-hole symmetry breaking as a result of hopping amplitudes beyond the nearest neighbors is possible in this case. The observation of the characteristic triangular features associated with quasi-bound states at H adatoms (in this work, Ref. [1] and many others) proves both the vacancy-like behavior of H adatoms and the absence of structural reconstruction. Modeling H adatoms in graphene by removing a site or placing an “infinitely” strong on-site potential is a standard description of H adatoms in graphene [4–6]. The differences between vacancies and H adatom in graphene have been discussed in Ref. [3] and numerous follow-up investigations.

Finally, following the Reviewer’s suggestion we have studied the case of a single vacancy in more detail using DFT. These results were presented in our previous reply. We have shown that vacancy with and without reconstruction displays a similar Kekulé vortex as the H adatom. In this revision, we go even further using the extraction method that we developed following the suggestion of Reviewer 1 (see reply to Reviewer 1). We extracted the sublattice A, sublattice B, and bond-centered contributions to intervalley scattering from the raw DFT data. These contributions are shown in Figs. S6-S7. The signal is qualitatively identical for all three defects (H adatom, unreconstructed vacancy and reconstructed vacancy): no winding on the impurity sublattice, a 4π -winding vortex on the complementary sublattice, and a 2π -winding vortex on the bond. This shows that a vacancy in graphene leads to the same Kekulé vortex as we observe for the H adatom.

We also would like to mention that, both our measurements and the scattering theory are not restricted to the topological zero mode of the BDI class. The Kekulé vortex can be observed at finite energy as well, which originates from the Berry phase of the Dirac cones of graphene.

We have added a discussion on the similarity between the Kekulé vortex around a H adatom and a realistic vacancy

in the revised version of the SI (Fig. S6-S7). We thank the Reviewer for this remark on a single vacancy that helped to strengthen the manuscript.

The authors do not write the Hamiltonian they use in their calculations (see SI part III), yet it seems to be the same Hamiltonian used in Ref. 23 by some of the co-authors of the present paper, a Hamiltonian which clearly breaks chiral symmetry (and particle-hole). As a result of particle-hole symmetry breaking, Friedel oscillations show up as thoroughly studied again in Ref.23. These oscillations impede the formation of a topological order (unlike a vacancy), and they prevent the existence of a true (i.e. topologically protected) zero mode.

We thank the Reviewer for noticing this omission. We now specify the explicit expression of the Hamiltonian in the revised version of the Supplementary Information.

Regarding the Friedel oscillations, we kindly disagree with the Reviewer. These oscillations at twice the Fermi wavevector do not rely on breaking particle-hole symmetry. For instance, such oscillations also appear for an infinite on-site potential, where particle-hole symmetry is restored. The Friedel oscillations rely instead on the existence of circular iso-energy contours at finite energy (i.e. away from the Dirac point degeneracy). Such oscillations at finite energy are also expected for a single vacancy. Furthermore, we also observe Friedel oscillations around the divacancy in our experiments, which should preserve sublattice symmetry in the Reviewer's description of vacancies. Thus, the Friedel oscillations we observe share similar features for both H adatoms and realistic, reconstructed vacancies. This is consistent with the similar Kekulé vortices they lead to, as discussed in point 3 above.

DFT based STM images displayed in Fig.2.b, present a relaxed atomic structure of graphene with an adatom. While being neat and convincing, these images must be consistent with experimental STM images presented in part I of the SI, which are identical to STM images in Fig.1.a of Ref.23 obtained in the very same setup which displays Friedel oscillations in a relaxed atomic structure. It seems therefore that there is a serious contradiction between raw STM images, numerics and calculations presented in parts I and II of the SI on one side and Fig.2.b in the main text based on DFT on the other side.

We would like to state that our data sets are consistent. The seeming contradiction may result from the fact that Fig. 2b is structural data, while the figures in the Supplementary Information are the electronic local density of states (LDOS). Hence, these data sets reveal different aspects of the system. First of all, as we have stated previously, experiments, T-matrix results, tight-binding and DFT (with and without relaxation) calculations all show the same patterns: no vortex in sublattice A, a 4π vortex in sublattice B and a 2π vortex in the bond-centered LDOS. These data sets are therefore very consistent with each other. As for the relaxation, we have simulated structural relaxation and LDOS using DFT in order to explore the interplay between the electronic states and lattice distortion. From the fact that both un-relaxed and relaxed structures show the same Kekulé texture, we conclude that such a phenomenon is not a consequence of lattice distortion. Therefore, together with our other theoretical studies this firmly establishes that the Kekulé vortex has its origin in the electronic structure of graphene. We draw the attention of the reviewer to the fact that the lattice distortions expected near a H adatom (as in Fig. 3) are of very small amplitude; too small to be resolved in STM. In Fig. 3b, we even had to artificially enhance these for the purpose of illustration (bond length is indicated by the thickness and color of the bonds). Indeed, the distortion is absolutely invisible to the naked eye without such an enhancement.

Moreover, the existence of an induced Kekule vortex is in contradiction with the breaking of particle-hole symmetry, a symmetry needed to observe any type of topological order in 2d in the presence of time reversal symmetry (according to the tenfold classification of topological insulators and superconductors, see e.g Ref.4). To give a firmer basis to their claim, the authors must answer these points and discuss in more details the atomic relaxation process which, they claim, seems to play a role.

We understand that the Kekulé vortex we observe can be different from the one expected within the topological classification for sublattice-symmetry models of the vacancy. However, our theoretical and experimental results show without any ambiguity that (i) the bond-centered contribution to the STM signal exhibits a Kekulé vortex, and (ii) both realistic vacancies and H adatoms lead to the same Kekulé vortex.

In conclusion, we would like to summarize our points:

- Following suggestion of the Reviewer, we add the Hamiltonian used in our work to our supplementary materials.

- We discussed the differences between H adatoms and vacancies as well as compared different modeling approaches.
- The relationship between DFT-simulated structures and STM images is discussed.

List of changes

- We implemented a quantitative method to extract and visualize the winding of the Kekulé bond parameter. The Fig. 1 of the main text is updated to present the order parameter. The details of the technique is described in the Supplementary Information, with the order parameter extracted for the H adatom (measurement) and vacancy (DFT).
- An extra figure (Fig. 2) is added to the main text, which includes the comparison between the Green's function results and the experimental results for the $\sqrt{3}$ signals.
- We reworded the final paragraph addressing the charge quantization.
- We add several references to recent works:
(Nature 620, 525–532 (2023); PRB 108, 054101 (2023) and arXiv:2307.05185)
- The Hamiltonian employed in the analysis is added to the Supplementary Information.

-
- [1] González-Herrero, H. *et al.* Atomic-scale control of graphene magnetism by using hydrogen atoms. *Science* **352**, 437–441 (2016).
- [2] Hou, C.-Y., Chamon, C. & Mudry, C. Electron Fractionalization in Two-Dimensional Graphenelike Structures. *Phys. Rev. Lett.* **98**, 186809 (2007).
- [3] Yazyev, O. V. & Helm, L. Defect-induced magnetism in graphene. *Phys. Rev. B* **75**, 125408 (2007). URL <https://link.aps.org/doi/10.1103/PhysRevB.75.125408>.
- [4] Soriano, D. *et al.* Magnetoresistance and Magnetic Ordering Fingerprints in Hydrogenated Graphene. *Phys. Rev. Lett.* **107**, 016602 (2011). URL <https://link.aps.org/doi/10.1103/PhysRevLett.107.016602>.
- [5] Soriano, D. *et al.* Spin transport in hydrogenated graphene. *2D Materials* **2**, 022002 (2015). URL <https://dx.doi.org/10.1088/2053-1583/2/2/022002>.
- [6] Balog, R. *et al.* Bandgap opening in graphene induced by patterned hydrogen adsorption. *Nature materials* **9**, 315–319 (2010).

Reviewers' Comments:

Reviewer #1:

Remarks to the Author:

I am pleased to say that the authors have sufficiently addressed all of my concerns, and I have no further comments. I can now clearly see the 2π phase winding when going around the adatom.

Reviewer #3:

Remarks to the Author:

The main points raised in my previous review focused on the differences between an adatom and a vacancy. My claim is that an adatom as opposed to a vacancy involves an additional energy scale (a new potential) which breaks particle-hole symmetry hence the possibility of topological order (or excitations) in a 2d systems with time reversal symmetry. The authors, in their response and in the new draft present a clear and tenable point of view that essentially the Hydrogen adatom they consider fulfils all requirements to (almost) preserve these two symmetries. This point of view is supported by state of the art STM images and numerical DFT numerics that must be seriously considered as was stated in a previous review.

The analysis of the winding around the Dirac point and the hint towards a demonstration of a Kekule vortex remain far fetched. Yet the added text about these issues leaves enough space for further discussions. This issue should not impede the publication of these neat and high quality experimental results. I therefore recommend the publication of this new version.

Reviewer #1 (Remarks to the Author):

I am pleased to say that the authors have sufficiently addressed all of my concerns, and I have no further comments. I can now clearly see the 2π phase winding when going around the adatom.

We thank the reviewer for supporting the publication of our manuscript

Reviewer #3 (Remarks to the Author):

The main points raised in my previous review focused on the differences between an adatom and a vacancy. My claim is that an adatom as opposed to a vacancy involves an additional energy scale (a new potential) which breaks particle-hole symmetry hence the possibility of topological order (or excitations) in a 2d systems with time reversal symmetry. The authors, in their response and in the new draft present a clear and tenable point of view that essentially the Hydrogen adatom they consider fulfils all requirements to (almost) preserve these two symmetries. This point of view is supported by state of the art STM images and numerical DFT numerics that must be seriously considered as was stated in a previous review.

The analysis of the winding around the Dirac point and the hint towards a demonstration of a Kekulé vortex remain far fetched. Yet the added text about these issues leaves enough space for further discussions. This issue should not impede the publication of these neat and high quality experimental results. I therefore recommend the publication of this new version.

We thank the reviewer for supporting the publication of our manuscript. In addition we have added a sentence to further clarify the particle hole symmetry braking which now leaves no room for ambiguity. In the penultimate paragraph of the section **Establishing the electronic origin of the Kekulé vortex**, we have added the following sentence:

However, this deviation is very small. According to first-principle calculations [34], the shift is approximately $t/16$, t is the nearest- neighbour hopping, which is just 0.01 of the total p_z bandwidth